# The Defect in Regulatory T Cells in Psoriasis and Therapeutic Approaches

**DOI:** 10.3390/jcm10173880

**Published:** 2021-08-29

**Authors:** Naoko Kanda, Toshihiko Hoashi, Hidehisa Saeki

**Affiliations:** 1Department of Dermatology, Nippon Medical School, Chiba Hokusoh Hospital, Inzai 270-1694, Japan; 2Department of Dermatology, Nippon Medical School, Tokyo 113-8602, Japan; t-hoashi@nms.ac.jp (T.H.); h-saeki@nms.ac.jp (H.S.)

**Keywords:** psoriasis, regulatory T cell, forkhead box protein 3, short chain fatty acid, butyrate, interleukin-17A, interleukin-23, dendritic cell, gut microbiome

## Abstract

Psoriasis is a chronic inflammatory skin disease characterized by accelerated tumor necrosis factor-α/interleukin (IL)-23/IL-17 axis. Patients with psoriasis manifest functional defects in CD4^+^CD25^+^ forkhead box protein 3 (Foxp3)^+^ regulatory T cells (Tregs), which suppress the excess immune response and mediate homeostasis. Defects in Tregs contribute to the pathogenesis of psoriasis and may attribute to enhanced inhibition and/or impaired stimulation of Tregs. IL-23 induces the conversion of Tregs into type 17 helper T (Th17) cells. IL-17A reduces transforming growth factor (TGF)-β1 production, Foxp3 expression, and suppresses Treg activity. Short-chain fatty acids (SCFAs), butyrate, propionate, and acetate are microbiota-derived fermentation products that promote Treg development and function by inducing *Foxp3* expression or inducing dendritic cells or intestinal epithelial cells to produce retinoic acids or TGF-β1, respectively. The gut microbiome of patients with psoriasis revealed reduced SCFA-producing bacteria, *Bacteroidetes,* and *Faecallibacterium*, which may contribute to the defect in Tregs. Therapeutic agents currently used, viz., anti-IL-23p19 or anti-IL-17A antibodies, retinoids, vitamin D3, dimethyl fumarate, narrow-band ultraviolet B, or those under development for psoriasis, viz., signal transducer and activator of transcription 3 inhibitors, butyrate, histone deacetylase inhibitors, and probiotics/prebiotics restore the defected Tregs. Thus, restoration of Tregs is a promising therapeutic target for psoriasis.

## 1. Introduction

Psoriasis is a chronic inflammatory skin disease characterized by accelerated tumor necrosis factor-α (TNF-α)/interleukin-23 (IL-23)/IL-17 axis and hyperproliferation and aberrant differentiation of epidermal keratinocytes (Figure 1) [1,2,3]. Dendritic cells (DCs) activated by various stimuli secrete TNF-α, which acts on themselves and induces their IL-23 secretion. IL-23 induces type 17 helper T (Th17) cells to proliferate and overproduce IL-17A and IL-22, which act on keratinocytes to promote their proliferation and production of the cytokines TNF-α and IL-17C; antimicrobial peptides or chemokines CXCL1/8 and CCL20 that recruit neutrophils, lymphocytes, and monocytes. Innate immune cells, such as type 3 innate lymphoid cells, γδT cells, or invariant natural killer T cells, also produce IL-17A, and are involved in the pathogenesis of psoriasis.

CD4^+^CD25^+^ forkhead box protein 3 (Foxp3)^+^ regulatory T cells (Tregs) suppress the excess immunity against various antigens, including self-antigens, and mediate self-tolerance and homeostasis. The transcription factor Foxp3 plays a central role in the development and function of Tregs [4]. In psoriasis, Tregs are functionally defective [5,6] and cannot sufficiently suppress the proliferation or inflammatory cytokine production of Th17 cells (Figure 1). Defects in Tregs may contribute to the development and exacerbation of psoriasis. Although Tregs increase in psoriatic lesional skin compared to healthy skin [7,8], the ratio of Th17 cells to Tregs is higher in the lesions. The results are conflicting regarding Treg frequency in the blood, with the frequency decreasing, no difference, or increasing in patients with psoriasis compared to that in healthy controls [8].

Natural Tregs are classified into thymus-derived Tregs (tTregs) and peripherally derived Tregs (pTregs). tTregs arise in the thymus and stably express Foxp3. pTregs arise in peripheral sites from conventional T cells in the presence of transforming growth factor (TGF)-β and IL-2 and by binding of self-antigens to the T-cell receptor (TCR) in combination with a co-stimulatory signal CD28. Foxp3 expression in pTregs is less stable than that in tTregs. Tregs can be generated in vitro from conventional T cells in the presence of TGF-β, and this population is known as in vitro-induced Tregs (iTregs). The demethylation level of Treg signature genes and stability of Foxp3 are lower in iTregs than pTregs [9]. In addition to Foxp3^+^ Tregs, CD4^+^ type 1 T regulatory (Tr1) cells represent another subset of Tregs defined by the expression of IL-10 and surface marker lymphocyte activation gene 3 and CD49b without Foxp3 and CD25 expression [10]. The relationship between Foxp3^+^ Tregs and Tr1 cells remains obscure, with both subsets employing common effector pathways, including IL-10, TGF-β, and cytotoxic T lymphocyte antigen 4 (CTLA4) [11].

The suppressive function of Tregs is mediated by multiple mechanisms (Figure 2) [12]; Tregs kill effector T cells or DCs through granzymes [13]; Tregs compete for IL-2 with effector T cells via CD25 and deprive IL-2, thus, inducing apoptosis [14,15]; Tregs release IL-10 and TGF-β, suppressing the proliferation of effector T cells [16,17], and the interaction of CTLA4 with CD80/86 downregulates CD80/86 on DCs and decreases their potency of antigen-presenting cells (APCs) to activate T cells [18]; interactions with Tregs induce the activity of indoleamine 2,3-dioxygenase in DCs, which in turn upregulates heme oxygenase-1 activity in Tregs, and secretion of its product carbon monoxide inhibits the proliferation of effector T cells [19].

The binding of Foxp3 to target genes induces the expression of *CD25*, *CTLA4*, *GITR*, or *HELIOS* while repressing the expression of *IL-2*, *IFN-**γ*, or *RORγt* in Tregs. *Foxp3* expression is mediated by five enhancer elements [4]: 5′-first conserved non-coding sequence (CNS) 0, promoter region, and three further CNS (CNS1–3); these elements are bound by specific transcription factors (Figure 3) [4]. CNS2 is called the Treg-specific demethylation region (TSDR), whose demethylated status confers the stability of Foxp3 expression [6]. In tTregs, CNS2 is fully demethylated and bound by full sets of transcription factors [6]. In pTregs, CNS2 is demethylated with lower stability, whereas in iTregs, this locus is rarely demethylated, making this subset very unstable [6]. *Foxp3* expression is initiated by the binding of self-antigens to TCR in combination with CD28 together with IL-2 and TGF-β [4]. The TCR/CD28 signal activates the binding of nuclear factor of activated T cells (NFAT), activator protein-1, forkhead box-containing protein O subfamily 1 (FOXO1), and nuclear receptor 4a to the promoter, NFAT binding to CNS1, cyclic AMP response element-binding protein binding to CNS2, and c-Rel binding to CNS3 [4]. TGF-β induces Smad2/3 binding to CNS1, whereas IL-2 induces signal transducer and activator of transcription (STAT) 5 binding to the promoter and CNS2 [4].

In this article, we review recent studies regarding the defect of Tregs in psoriasis and the therapeutic agents that restore the defective Tregs. The defect in Tregs in psoriasis may attribute to enhanced inhibition and/or impaired stimulation of the generation and maintenance of Tregs. Restoration of Tregs is necessary for the control of psoriasis, and is a promising therapeutic target.

## 2. Genetic and Epigenetic Evidence for Defected Tregs in Psoriasis

Gao et al. reported that Chinese patients with psoriasis showed polymorphisms in *Foxp3* [20]. The *Foxp3*-3279 AC and IVS9+459 GG genotypes were associated with an increased risk of psoriasis in a Chinese population, indicating that they may increase the risk for psoriasis by quantitatively and functionally influencing Tregs. Larger population-based studies are needed to confirm the universality of these findings.

Ngalamilka et al. reported that the peripheral blood of patients with psoriasis had significantly higher methylation levels of *Foxp3* TSDR compared to healthy controls [21]. TSDR hypermethylation is associated with chromatin condensation and downregulation of *Foxp3* expression; therefore, the results indicate downregulation of *Foxp3* and reduction of Tregs in psoriasis patients’ blood.

## 3. Downregulation of Tregs by Cytokines or Mediators Which Are Upregulated in Psoriasis

CD4^+^CD25^+^Foxp3^+^ Tregs in psoriatic skin lesions can be converted to retinoic acid (RA) receptor-related orphan nuclear receptor γt (RORγt)^+^IL-17A-producing cells, and the conversion is promoted by IL-23 [22]. IL-23 reduces Foxp3 expression in Tregs [22]. IL-23 is produced by APCs and keratinocytes in psoriatic skin lesions [23], and keratinocyte-derived IL-23 may contribute to the downregulation of Tregs.

IL-17A acts on human Tregs and reduces their suppressive activity on effector T cell proliferation, suppresses Foxp3 and TGF-β expression, and enhances IFN-γ and T-bet expression [24]. These results indicate a downregulatory role of IL-17A in Tregs. However, contradictory results have been reported. IL-17A knockout mice failed to induce Tregs as efficiently as wild-type mice with autoimmune uveitis [25], and blocking IL-17A abolished the CD4^+^CD25^+^Treg function required for preventing corneal allograft rejection [26]; thus, indicating that IL-17A is required for the induction and/or maintenance of Tregs in the eyes. Either up- or down-regulation by IL-17A in Tregs may depend on the target organs or species, and further studies are needed for clarification.

Yang et al. reported that Tregs from the peripheral blood of patients with psoriasis produced IFN-γ, TNF-α, and IL-17A, together with enhanced phosphorylation of STAT3 and impaired suppressive functions [27]. STAT3 inhibitor Stattic V restrained IFN-γ, TNF-α, and IL-17A expression and restored the suppressive function of Tregs in vitro [27]. IL-6, IL-21, and IL-23 induced the phosphorylation of STAT3 in Tregs in vitro [27]. These findings indicate that IL-6, IL-21, and IL-23 cytokines whose expressions are elevated in psoriatic lesions, may impair the suppressive function of Tregs and induce the conversion of Tregs into Th1/Th17 cells via STAT3 phosphorylation. IL-6 and IL-21 also render effector T cells refractory to Treg-mediated suppression via the STAT3 pathway [28,29].

Akt-induced phosphorylation and inactivation of the transcription factor FOXO1 is another mechanism underlying Treg dysfunction in psoriasis [30]. Circulating Tregs in patients with psoriasis expressed high levels of T-bet and IFN-γ mRNAs, showing a Th1-like phenotype in addition to enhanced phosphorylation of FOXO1 and Akt and cytoplasmic localization of FOXO1. FOXO1 can bind to the promoter of TBX21, which codes T-bet, to inhibit its expression, whereas Akt-induced phosphorylation of FOXO1 induces its cytoplasmic translocation from the nucleus and impairs its transrepressive activity. Serum from patients with psoriasis induced the activation of Akt and phosphorylation and cytoplasmic translocation of FOXO1 in Tregs from healthy controls in vitro, although the Akt-inducing molecules in the serum were not identified.

The expression of microRNA-210 is increased in circulating CD4^+^ T cells from patients with psoriasis, and this increase may contribute to the reduced Foxp3 mRNA and protein levels in the patients’ CD4^+^ T cells [31]. microRNA-210 binds to the 3′-untranslated region of *Foxp3*, and the overexpression of microRNA-210 inhibits Foxp3 expression in CD4^+^ T cells from healthy controls, whereas inhibition of microRNA-210 increases Foxp3 expression in CD4^+^ T cells from patients with psoriasis.

## 4. Stimulators for Tregs and Their Impairment in Psoriasis

Short-chain fatty acids (SCFAs), such as butyrate, propionate, and acetate, are derived from gut microbial fermentation of dietary fiber, and promote the generation and function of Tregs in the gut and systemically [32]. Skin commensals, such as *Cutibacterium acnes*, also produce SCFAs [33,34], which may stimulate Tregs in the skin. Butyrate acts on DCs and induces the expression of retinaldehyde dehydrogenase, an enzyme that synthesizes RAs [35,36] that promote *Foxp3* expression in pTregs. Butyrate acts as an inhibitor of histone deacetylase (HDAC) and induces histone H3 acetylation on *Foxp3* intronic enhancer, allowing the expression of *Foxp3* in naïve CD4^+^ T cells, inducing their differentiation into pTregs. SCFAs induce intestinal epithelial cells to produce TGF-β [37], which can contribute to *de novo* generation of pTregs by inducing *Foxp3* expression via Smad2/3.

Acetate, propionate, and butyrate stimulate the proliferation of tTregs via cell surface GPR43 [38]. Butyrate binds GPR41 on medullary thymic epithelial cells, and promotes the expression of a transcription factor autoimmune regulator (AIRE), which induces the generation of tTregs [39]. Acetate also induces AIRE expression in cortical thymic epithelial cells [40].

It has been reported that the gut microbiome of patients with psoriasis and psoriatic arthritis showed a decrease in SCFA-producing bacteria, including *Bacteroidetes*, *Prevotella*, *Akkermansia*, *Faecalibacterium*, and *Ruminococcus* [41,42]. In particular, a decrease in *Faecalibacterium prausnitzi* and *Akkermansia muciniphila*, which produce butyrate, was noted in patients with psoriasis and psoriatic arthritis [43,44]. The gut microbiome of patients with psoriasis is characterized by an increase in the phylum *Firmicutes* and a decrease in the phylum *Bacteroidetes* [45], which is related to an impaired gut epithelial barrier and reduced butyrate production. Shapiro et al. reported that genes encoding butyrate kinase and phosphate butyryltransferase, enzymes involved in butyrate synthesis, were present in lower proportions in the feces of patients with psoriasis than in the control cohort [46]. The alteration of the gut microbiome in patients with psoriasis may contribute to the defect of Tregs via the reduction of SCFAs.

Vitamin D3, obtained via dietary intake or synthesis in the skin by sun exposure, is metabolized to its active form, 1,25-dihydroxyvitamin D3. 1,25-Dihydroxyvitamin D3 binds to the vitamin D receptor (VDR) that heterodimerizes with the retinoid X receptor (RXR); the heterodimer binds to the vitamin D response element on the *Foxp3* enhancer CNS1 in naïve T cells, leading to *Foxp3* expression and generation of pTregs [4]. It has been reported that serum 25-hydroxyvitamin D3 levels are reduced in patients with psoriasis or psoriatic arthritis compared to controls [47] and that the expression of VDR is decreased in psoriatic skin lesions [48]. The reduced levels of vitamin D3 and/or VDR may be related to defects in Tregs in psoriasis via the reduction of *Foxp3* expression.

## 5. Therapeutic Approach to Restore the Defect of Tregs in Psoriasis

Restoration of defective Tregs is a promising therapeutic target for psoriasis and psoriatic arthritis. Therapeutic modalities that potentiate Tregs should be selected for a patient population with defective Tregs. For such selection, specific and convenient methods of testing Treg numbers and functions should be developed in the near future.

### 5.1. Therapeutic Agents Currently Used for Psoriasis

Several biologics or low-molecular-weight agents currently used for psoriasis can restore defective Tregs in psoriasis (Table 1).

#### 5.1.1. Biologics

Anti-IL-23p19 antibodies, risankizumab, guselkumab, tralokinumab, or anti-IL-12/23p40 antibody ustekinumab may reverse the IL-23-induced conversion of Tregs into pathogenic Th17 cells and may increase the number and/or suppressive function of Tregs in psoriasis. Anti-IL-23p19 antibody treatment in an imiquimod-induced psoriasis mouse model increased the number of Foxp3^+^ cells in the lesions, and adoptive transfer of Tregs from anti-IL-23p19 antibody-treated mice improved psoriasis-like dermatitis in the donor mice [49].

Anti-IL-17A antibodies, secukinumab and ixekizumab, or the anti-IL-17RA antibody brodalumab may counteract IL-17A-induced impairment of Tregs. Treatment with secukinumab in patients with psoriasis restored the suppressive function and increased TGF-β production in Tregs, as well as reduced the psoriasis area and severity index (PASI) [24].

#### 5.1.2. Vitamin D3

Topical treatment with maxacalcitol, vitamin D3, in imiquimod-induced psoriasis mice model increased Treg infiltration and IL-10 expression in skin lesions, and adoptive transfer of Tregs from maxacalcitol-treated mice ameliorated psoriasis-like dermatitis in donor mice, indicating a functional suppressive phenotype [50]. Several randomized controlled trials (RCTs) have reported that systemic vitamin D supplementation reduces PASI scores [32,51]. However, the effects have not been verified by a systematic review and meta-analysis of RCTs [52]. More RCTs with larger sample sizes are needed to produce robust results.

#### 5.1.3. Retinoids

Synthetic vitamin A derivatives, retinoids, such as etretinate or acitretin, are absorbed in the small intestine and delivered to the fat, liver, gut, or kidney, where they are metabolized to the active acid form RAs [53]. RA binds to the retinoic acid receptor (RAR), which heterodimerizes with RXR, and the heterodimer binds the RA response element on CNS1 of *Foxp3*, inducing *Foxp3* expression and the generation of pTregs from naïve T cells.

#### 5.1.4. Narrow-Band Ultraviolet (UV) B Therapy

Narrow-band UVB therapy increased Treg number and Foxp3 mRNA levels in peripheral blood mononuclear cells of patients with psoriasis and reduced Th1/Th17 cells [54]. Moreover, narrow-band UVB treatment on keratinocytes upregulates the expression of receptors of activated nuclear factor-κB ligand (RANKL), which interacts with RANK on DCs, promoting DCs to expand the number of Tregs systemically [55,56].

#### 5.1.5. Dimethyl Fumarate (DMF)

The European Medicines Agency approved an oral formulation of DMF for the treatment of moderate-to-severe psoriatic plaque in adults in 2017 [57]. DMF and its active metabolite monomethylfumarate downregulate inflammatory cytokine production in T cells and induce the shift from Th1/Th17 to Th2 by modulation of intracellular glutathione levels and, ultimately, cellular responses to oxidative stress; hence, modulating the activity of the transcription factors nuclear factor-erythroid 2-related factor 2, NF-κB, and hypoxia-inducible factor 1-α or binding to cell surface hydroxyl–carboxylic acid receptor 2 [58].

Oral DMF treatment in Lewis rats increased Foxp3 mRNA levels in the ileum and CD4^+^CD25^+^ Tregs in Peyer’s patches [59]. The adaptive transfer of these Tregs effectively improved experimental autoimmune neuritis in recipient rats [59]. The upregulation of Tregs by DMF may be mediated by SCFAs as DMF treatment increased the number of SCFAs-producing bacteria, such as *Gemella*, *Roseburia*, *Bacillus*, and *Bacteroides* in the gut [60].

DMF treatment increased Treg frequency and decreased Th17 cells in patients with psoriasis [61]. In vitro DMF treatment induced oxidative stress, which reduced the viability and proliferation of CD4^+^CD25^−^ conventional T cells but did not reduce Tregs [62], by virtue of the increased expression of cell surface-reduced thiols [61] or thioredoxin-1 [63], protecting Tregs from oxidative stress. The oxidative effects of DMF may favor Tregs relative to Th17 cells, which may be another mechanism for the anti-psoriatic effects of DMF.

#### 5.1.6. Janus Kinase (JAK) Inhibitors

The JAK family of non-receptor tyrosine kinases transduce signals from a multitude of cytokines [64]. The binding of cytokines to their receptors enables the activation of receptor-associated JAK and JAK-induced phosphorylation of receptors, allowing STAT to bind to receptors and to be phosphorylated by JAKs, leading to the dimerization of STATs, nuclear translocation, and transcription of target genes. JAK inhibitors suppress the JAK/STAT signaling pathways and, thus, block the effects of inflammatory cytokines, such as IL-6, IFN-γ, IL-22, and IL-21, which are involved in the pathogenesis of psoriasis. The JAK 1/3 inhibitor tofacitinib is approved by the Food and Drug Administration for the treatment of psoriatic arthritis [64]. The JAK1 inhibitor upadacitinib is approved in Japan for psoriatic arthritis. To date, the effects of JAK inhibitors on the number or function of Tregs in patients with psoriasis, or psoriasis mice models, have not been reported. However, tofacitinib increased the number of Tregs and reduced the number of Th17 cells in the liver and spleen of mice with concanavalin A-induced hepatitis [65]. Although the precise mechanism is unknown, it has been reported that the suppressive capacity of Tregs on effector T cells is resistant to the blocking effects of tofacitinib, whereas the function of effector T cells is more sensitive to these effects [66].

### 5.2. Therapeutic Agents under Development

Therapeutic agents under development that promote the generation and function of Tregs for psoriasis treatment are mentioned in Table 1. Patients resistant to or with insufficient response to current therapy modes, such as systemic immunosuppressive medicine, may include a population with prominent defects in Tregs. For such a population, agents promoting Tregs may complement the therapeutic effects of the current therapy.

#### 5.2.1. SCFAs

Ex vivo treatment of psoriatic lesional skin with sodium butyrate restored the reduced Treg number and IL-10 and Foxp3 expression and normalized the enhanced expression of IL-17A and IL-6 [67]. Topical application of sodium butyrate to imiquimod-induced psoriasis-like dermatitis in mice increased *IL-10* and *Foxp3* transcripts and reduced inflammation and *IL-17A* transcripts [67]. The beneficial effects of sodium butyrate are abolished by the depletion of Tregs [67]. Topical SCFAs may be a promising therapy for psoriasis.

#### 5.2.2. STAT3 Inhibitors

Topical treatment with STA-21, a STAT3 inhibitor, improved human psoriatic skin lesions as well as psoriasis-like dermatitis in K5.Stat3C transgenic mice, indicating a promising role of this agent in the treatment of psoriasis [68]. STA-21 increased the number and function of CD4^+^CD25^+^Foxp3^+^ Tregs in IL1-receptor α knockout mice, a model for rheumatoid arthritis [69]. Adoptive transfer of Tregs from STA-21-treated mice markedly suppressed inflammatory arthritis, and in vitro treatment with STA-21 increased Foxp3 mRNA levels in human and murine CD4^+^ T cells [69]. STA-21 increased the level and phosphorylation of STAT5, a critical transcription factor of *Foxp3* expression, in addition to a reduction in STAT3 levels and phosphorylation [69]. The reciprocal regulation of STAT3 and STAT5 by STA-21 may increase *Foxp3* expression and the suppressive function of Tregs. In vitro treatment of murine splenocytes with STA-21 also increased the frequency of CD4^+^CD25^+^Foxp3^+^ Tregs [70]. Stattic V, another STAT3 inhibitor, restored the suppressive function of Tregs in patients with psoriasis [27]. STAT3 inhibitors, including STA-21, may improve psoriasis by potentiating Tregs.

#### 5.2.3. Probiotics/Prebiotics

Probiotics are living microorganisms that confer health benefits to the host when administered in adequate amounts [71]. Most microorganism probiotics belong to the lactic acid-producing genera *Lactobacillus* and *Bifidobacterium*. Oral administration of these genera increased Foxp3^+^ Treg responses in dextran sulfate sodium-induced colitis [72] or experimental autoimmune encephalomyelitis [73] in mice as well as ameliorated inflammation. Prebiotics are non-digestible fructooligosaccharides, inulins, or galactooligosaccharides that stimulate the growth of beneficial bacteria, such as *Bifidobacterium* [74]. Inulin, a soluble dietary fiber, is fermented by the gut microbiome to generate SCFAs. Oral administration of inulin in rats altered the composition of the gut microbiome and increased the probiotic bacteria *Lactobacillus* and SCFA-producing bacteria *Lachnospiraceae*, *Phascolarctobacterium*, and *Bacteroides* [75]. Female mice fed an inulin-enriched diet during pregnancy and lactation showed an abundance of *Bacteroides* in the gut microbiome and increased plasma SCFA levels, and their offspring had increased frequencies of tTregs and pTregs and increased expression of AIRE in the thymus [39].

Oral administration of *B. infantis* 35624 in patients with psoriasis reduced plasma levels of TNF-α and CRP [76]. Oral administration of *B. longum* CECT 7347, *B. lactis* CECT 8145, and *L. rhamnosus* CECT 8361 in patients with psoriasis resulted in higher PASI 75 compared to the placebo group [77] with an increase in the probiotic bacteria *Collinsella* and *Lactobacillus* in the gut. Feeding a diet rich in fucoidan (a seaweed fiber) in the psoriasis mice model, induced by a *Traf3ip2* mutation, ameliorated symptoms of psoriasis-like dermatitis with increasing *Bacteroides* in the gut [78]. However, whether these probiotic/prebiotic treatments improve the number or function of Tregs has not been examined in patients with psoriasis or mice models and should be further verified. Future studies using synbiotics, probiotics combined with prebiotics, might be promising.

#### 5.2.4. HDAC Inhibitors

The enhancer elements of *Foxp3* are bound by histones, and histone acetylation of these elements allows the gene to be accessible for transcription factors and RNA polymerase, promoting gene expression [4]. HDAC inhibitors enhance histone acetylation of these elements and induce *Foxp3* expression [4]. Foxp3 is also regulated post-transcriptionally. Foxp3 protein associates with histone acetylase and HDAC, and acetylation of this protein increases its stability, whereas its deacetylation makes it susceptible to proteasomal degradation [4]. HDAC inhibitors may, thus, stabilize Foxp3 protein. The pan-HDAC inhibitor trichostatin A acts on human peripheral blood-derived CD4^+^CD25^+^Foxp3^+^ Tregs, reversing their conversion into Th17 cells and increasing *Foxp3* expression in healthy controls [79] and in patients with psoriasis [22]. These results indicate a promising role for HDAC inhibitors in psoriasis treatment. However, HDAC inhibitors have not been examined as therapeutic agents other than in oncology. Thus far, 18 HDAC enzymes have been identified; 11 are Zn^2+^-dependent (HDAC 1–11), and seven require nicotinamide adenine dinucleotide (Sirt 1–7). Among these, Tregs express HDAC 1–11 and Sirt 1–4. Individual HDAC isoforms differentially regulate the transcription and/or stabilization of Foxp3 [4]; thus, specific inhibitors for individual HDACs should be examined for their ability to generate and/or maintain Tregs.

## 6. Conclusions

Patients with psoriasis are associated with the impaired function of Tregs and disturbed Treg/Th17 balance, which may contribute to the development and exacerbation of this disease. The defect in Tregs in psoriasis may be attributable to enhanced inhibition and/or impaired stimulation of the generation and maintenance of Tregs. Tregs can convert into Th17- or Th1-like phenotypes in patients with psoriasis. Cytokines or mediators that are upregulated in psoriasis, such as IL-23, IL-17A, IL-6, and IL-21, may induce the conversion of Tregs into Th17/Th1 cells. SCFAs stimulate the induction of Tregs via *Foxp3* expression or through their action on DCs or intestinal epithelial cells. The reduction of SCFA-producing bacteria, such as *Bacteroides*, in the gut microbiome of patients with psoriasis, may contribute to the defect in Tregs. Several therapeutic modalities currently used for psoriasis treatment, such as anti-IL-23p19 or anti-IL-17A antibodies, retinoids, topical vitamin D3, DMF, and narrow-band UVB, can restore defective Tregs. Agents that potentiate Tregs, such as STAT3 inhibitors, butyrate, HDAC inhibitors, or probiotics/prebiotics, are under development for the treatment of psoriasis. Restoration of defective Tregs may be a promising therapeutic target for psoriasis.

## Figures and Tables

**Figure 1 jcm-10-03880-f001:**
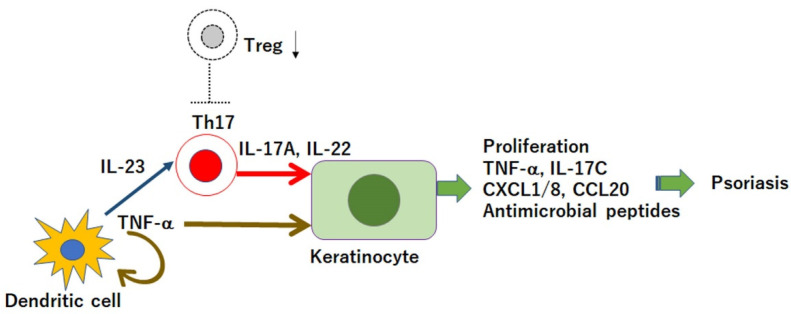
Tumor necrosis factor (TNF)-α/interleukin (IL)-23/IL-17 axis, and the defect in regulatory T cells (Tregs) in the pathogenesis of psoriasis. Dendritic cells secrete TNF-α that induces IL-23 secretion. IL-23 induces type 17 helper T (Th17) cells to produce IL-17A and IL-22, which act on keratinocytes and promote the proliferation and production of cytokines, antimicrobial peptides, or chemokines; thus, inducing inflammation. In psoriasis, Tregs are dysfunctional and cannot sufficiently suppress the activity of Th17 cells.

**Figure 2 jcm-10-03880-f002:**
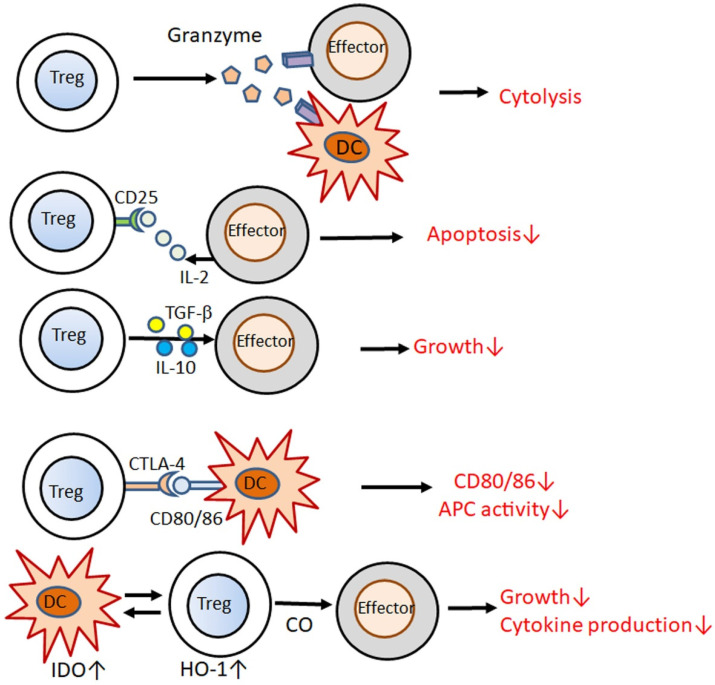
Possible mechanisms for immunosuppression by CD4^+^CD25^+^ Forkhead box protein 3 (Foxp3)^+^ regulatory T cells (Tregs). Tregs kill effector T cells and dendritic cells (DCs) by granzymes; Tregs deprive interleukin (IL)-2 from effector T cells; Tregs secrete IL-10 and transforming growth factor-β (TGF-β), which suppress the proliferation of effector T cells; cytotoxic T-lymphocyte antigen 4 (CTLA4) binding to CD80/86 downregulates CD80/86 and capacity of antigen-presenting cells (APCs) in DCs; interaction with Tregs induces the activity of indoleamine 2,3-dioxygenase (IDO) in DCs, which further activates heme-oxygenase-1 (HO-1) in Tregs, releasing carbon monoxide (CO), which inhibits the activity of effector T cells.

**Figure 3 jcm-10-03880-f003:**
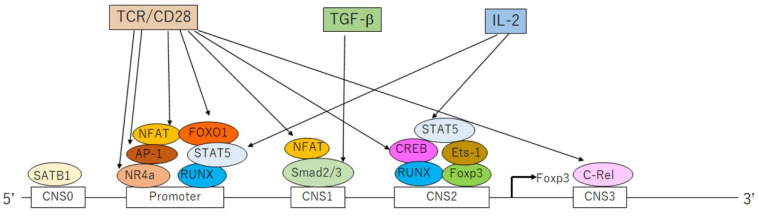
*Forkhead box protein 3* (*Foxp3*) gene expression in regulatory T cells. *Foxp3* gene contains five enhancer elements: four conserved non-coding sequences (CNS0–3) and a promoter. *Foxp3* expression is initiated by the binding of self-antigens to the T-cell receptor (TCR) in combination with a co-stimulatory signal, CD28, together with transforming growth factor (TGF)-β and interleukin (IL)-2. These activation signals provoke the recruitment and binding of transcription factors to individual response elements. The TCR/CD28 signal induces the binding of activator protein-1 (AP-1), nuclear factor of activated T cells (NFAT), nuclear receptor 4a (NR4a), forkhead box-containing protein O subfamily 1 (FOXO1), cyclic AMP response element-binding protein (CREB), and c-Rel. TGF-β induces the binding of Smad2/3, and IL-2 induces the binding of signal transducer and activator of transcription 5 (STAT5). SATB1, special AT-rich sequence-binding protein 1; RUNX, runt-related transcription factor.

**Table 1 jcm-10-03880-t001:** Therapeutic Agents Restoring Defected Regulatory T Cells (Tregs) in Psoriasis.

Therapeutic Agents	Mechanisms for Restoring Tregs
**Currently Used**	
Anti-IL-23p19 or anti-IL-12/23p40 antibodies	Reversing conversion from Tregs into Th17 cells
Anti-IL-17A or anti-IL-17RA antibodies	Increasing TGF-β secretion, Foxp3 expression, and suppressive function of Tregs
Topical vitamin D3	Increasing Foxp3 expression through VDR
Retinoids	Increasing Foxp3 expression through RAR
Narrow-band UVB	Increasing RANKL expression in keratinocytes and inducing DCs to expand Tregs
Dimethyl fumarate	Increasing the frequency of Tregs resistant to dimethyl fumarate-induced oxidative stress or increasing SCFA-producing bacteria in the gut
**Under development**	
SCFAs	Increasing Foxp3 expression via HDAC inhibition, TGF-β production in IEC, RA synthesis in DCs, and proliferation of tTregs
STAT3 inhibitors	Increasing Foxp3 expression via induction of STAT5
Probiotics	Increasing SCFA production in the gut
Prebiotics	Increasing SCFA-producing bacteria in the gut
HDAC inhibitors	Increasing Foxp3 expression and stabilization

IL, interleukin; Th17, type 17 helper T; SCFA, short-chain fatty acid; HDAC, histone deacetylase; STAT, signal transducer and activator of transcription; Foxp3, Forkhead box protein 3; DC, dendritic cell; IEC, intestinal epithelial cell; RANKL, receptor activator of the nuclear factor-κB ligand; RAR, retinoic acid receptor; VDR, vitamin D receptor; TGF, transforming growth factor; tTreg, thymus-derived Treg; UV, ultraviolet.

## Data Availability

Not applicable.

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
