# Peer review of "The Defect in Regulatory T Cells in Psoriasis and Therapeutic Approaches"

_jcm, 2021, doi:10.3390/jcm10173880_

Round 1

Reviewer 1 Report

Line 263 - spelling - should be kinase not kinse

Reviewer 2 Report

This revised manuscript entitled to ‘The defect in regulatory T cells in psoriasis and therapeutic approaches (Manuscript ID: jcm-1357236)” well-reflected what I commented before, This review manuscript would be helpful for understanding the function of regulatory T cells’ in psoriasis and development anti-psoriasis agents. Thus, I recommend this review article Accept.

This manuscript is a resubmission of an earlier submission. The following is a list of the peer review reports and author responses from that submission.

Round 1

Reviewer 1 Report

The authors undertook an important subject of the role of regulatory T cells in psoriasis and how the current and future (under development) treatments may act via the mechanisms involving Treg cells. The paper is clearly divided into sections, the figures clarifying the role of Tregs in psoriasis. The table summarizes the therapies and their mechanisms.

Questions/comments

  1.  

Lines 49-55 – please, clarify what is the relationship between pTregs and iTregs if any? It is not clear from the fragment.

  1.  

Tregs abnormalities in blood in psoriatic patients are well described in the paper, however there is lack of paragraph describing data from the literature regarding the psoriatic skin Tregs – please, have a look at the review papers below which can take you to the original articles and consider adding a paragraph/fragment on this topic.

https://doi.org/10.1111/bjd.19380

https://doi.org/10.1093/intimm/dxz020

  1.  

Table 1

- Narrow-band UVB - Increasing vitamin D3 synthesis in the skin

And lines

231-234 Narrow-band UVB therapy increased Treg number and Foxp3 mRNA levels in peripheral blood mononuclear cells of patients with psoriasis as well as the reduction of Th1/Th17 cells [50]. The mechanism for the restoration of Tregs is unknown; however, the restoration might partially be attributable to the synthesis of vitamin D3 induced by narrow-band UVB in the skin.

In the table the authors are stating that vitamin D3 synthesis in the skin is responsible for restoring Tregs while in the fragment of text it is only partially attributable with a different mechanism suggested (blood). Literature regarding phototerapy mechanisms and Tregs in the skin is available, it may be worth adding the possible skin mechanism involving Tregs to the narrow-band UVB paragraph and table 1.

  1.  

Vitamin D3 – lines 219-223. Several clinical studies reported the improvement of psoriasis by systemic vitamin D supplementation [29] [48].

It may be worth to rephrase. Please, have a look at the meta-analysis: Theodoridis X et al. Effectiveness of oral vitamin D supplementation in lessening disease severity among patients with psoriasis: A systematic review and meta-analysis of randomized controlled trials. Nutrition. 2021 Feb;82:111024. doi: 10.1016/j.nut.2020.111024.

Moreover, vitamin D3 is not a registered treatment for psoriasis. Table 1 should be adjusted as well.

  1.  

Conclusions 323-324

Patients with psoriasis are associated with the decrease of number and/or function of Tregs, which may contribute to the development and exacerbation of this disease

The sentence needs to be rephrased grammatically but also in majority of papers regarding Tregs in psoriatic lesions the number is increased – please see point 2.

  1. Is there any knowledge on the possible influence of JAK/STAT pathway inhibitors eg tofacitinib and baricitinib (already registered for psoriasis) on Tregs?

Author Response

Thank you for reviewing our manuscript and important comments. The responses to your comments are described as follows:

Comment 1

Lines 49-55 – please, clarify what is the relationship between pTregs and iTregs if any? It is not clear from the fragment.

Response: pTregs are in vivo naturally generated in peripheral sites from conventional T cells while iTregs are induced in vitro from conventional T cells. The demethylation level of Treg signature genes and stability of Foxp3 are lower in iTregs compared to those in pTregs (lines 53-59). A relevant paper is newly cited (ref 9).

Comment 2 

Tregs abnormalities in blood in psoriatic patients are well described in the paper, however there is lack of paragraph describing data from the literature regarding the psoriatic skin Tregs – please, have a look at the review papers below which can take you to the original articles and consider adding a paragraph/fragment on this topic.

https://doi.org/10.1111/bjd.19380

https://doi.org/10.1093/intimm/dxz020

Response: We have added the sentences showing the increased Treg number in psoriasis lesional skin (lines 49-52) with citation of the papers suggested by the reviewer (refs 7,8). 

Comment 3 

Table 1- Narrow-band UVB - Increasing vitamin D3 synthesis in the skin And lines 231-234 Narrow-band UVB therapy increased Treg number and Foxp3 mRNA levels in peripheral blood mononuclear cells of patients with psoriasis as well as the reduction of Th1/Th17 cells [50]. The mechanism for the restoration of Tregs is unknown; however, the restoration might partially be attributable to the synthesis of vitamin D3 induced by narrow-band UVB in the skin.

In the table the authors are stating that vitamin D3 synthesis in the skin is responsible for restoring Tregs while in the fragment of text it is only partially attributable with a different mechanism suggested (blood). Literature regarding phototherapy mechanisms and Tregs in the skin is available, it may be worth adding the possible skin mechanism involving Tregs to the narrow-band UVB paragraph and table 1.

Response: The mechanism for narrow-band UVB-induced Treg induction is that UVB upregulates RANKL expression of keratinocytes, which induces dendritic cells to expand Tregs via RANKL/RANK interaction (lines 240-243). The relevant papers are newly cited (refs 55,56) and the description in table 1 is accordingly revised.

Comment 4 

Vitamin D3 – lines 219-223. Several clinical studies reported the improvement of psoriasis by systemic vitamin D supplementation [29] [48].

It may be worth to rephrase. Please, have a look at the meta-analysis: Theodoridis X et al. Effectiveness of oral vitamin D supplementation in lessening disease severity among patients with psoriasis: A systematic review and meta-analysis of randomized controlled trials. Nutrition. 2021 Feb;82:111024. doi: 10.1016/j.nut.2020.111024. 

Moreover, vitamin D3 is not a registered treatment for psoriasis. Table 1 should be adjusted as well.

Response: Though several randomized control trials (RCTs) revealed the reduction of psoriasis area and severity index (PASI) by systemic vitamin D supplementation, the effectiveness was not verified by systematic review and meta-analysis of RCTs, and RCTs with larger sample sizes are required for its verification. The text is revised (lines 228-231) and the paper showing meta-analysis of RCTs is newly cited (ref 52). Systemic vitamin D3 administration is not a registered treatment for psoriasis while topical vitamin D3 is the standard treatment for psoriasis. The description in Table 1 is thus revised as ‘topical vitamin D3.’

Comment 5 

Conclusions 323-324 Patients with psoriasis are associated with the decrease of number and/or function of Tregs, which may contribute to the development and exacerbation of this disease

The sentence needs to be rephrased grammatically but also in majority of papers regarding Tregs in psoriatic lesions the number is increased – please see point 2.

Response: Treg number is rather increased in psoriasis skin lesions compared to healthy skin, and the text is thus revised in introduction (lines 49-52) and conclusion (line 347). This manuscript is wholly revised by native English writers.

Comment 6

Is there any knowledge on the possible influence of JAK/STAT pathway inhibitors eg tofacitinib and baricitinib (already registered for psoriasis) on Tregs?

Response: we have added the column describing the effects of JAK inhibitors on Tregs (lines 263-278) with a relevant paper newly cited (ref 64). To date, the direct effects of JAK inhibitors on Tregs in psoriasis patients or psoriasis mice model have not been reported. However, tofacitinib-induced increase of Treg number is reported in concanavalin A-induced hepatitis (ref 65). The mechanism for that effect is not shown, however, one study reports that the suppressive activity of Tregs on effector T cells is resistant to blocking effects of tofacitinib while the function of effector T cells is sensitive to those effects above (ref 66).

Reviewer 2 Report

I have no comments to the authors.

Author Response

Thank you for reviewing our manuscript. The manuscript has been edited by native English writers.

Reviewer 3 Report

Minor

  1. This manuscript should be reviewed by native speaker. Several sentences/paragraphs are hard to understand.
  2. Figure legends: Each ‘Figure legends’ are overlapped with main text of the manuscript. The authors should rewrite ‘Figure legends’.

Author Response

Thank you for reviewing our manuscript and important comments. The responses to your comments are described as follows:

Minor comments

  1. This manuscript should be reviewed by native speaker. Several sentences/paragraphs are hard to understand.

Response: The manuscript has been revised by native English writers. The corrected portions are highlighted.

  1. Figure legends: Each ‘Figure legends’ are overlapped with main text of the manuscript. The authors should rewrite ‘Figure legends’.

Response: We have rewritten the legends. The correction is highlighted for each figure legend.